# Wave Height Attenuation over a Nature-Based Breakwater of Floating Emergent Vegetation

**Yanhong Li** [1,*], **Dongliang Zhao** [2], **Guoliang Yu** [1,*] and **Liquan Xie** [3]

1   KLMIES, MOE, State Key Laboratory of Ocean Engineering, School of Naval Architecture,
    Ocean and Civil Engineering, Shanghai Jiaotong University, Shanghai 200240, China
2   CCCC Second Harbor Engineering Company Ltd., Wuhan 430040, China
3   Department of Hydraulic Engineering, Tongji University, Shanghai 200092, China; xie_liquan@tongji.edu.cn
*   Correspondence: yyhli@sjtu.edu.cn (Y.L.); yugl@sjtu.edu.cn (G.Y.)

**Abstract:** The nature-based breakwater of floating emergent vegetation (BFEV) provides protection for water banks and various engineering structures from wave erosion. Compared with the convenient hard breakwater, the BFEV is beneficial to the resilient and sustainable development of rivers, lakes, coasts, and marine areas because it is free of new pollution. As a new breakwater, the unrevealed effect and efficiency of the BFEV on wave attenuation are to be investigated through a set of 312 physical tests in a rectangular indoor water flume in the present study. Results show that the wave height attenuates by 38–62%. Based on statistical methods, the main influencing factors of the wave transmitted coefficient ($C_t$) are found to be closely dependent on three conventional and newly proposed dimensionless parameters ($\lambda_1$, $\lambda_2$, $\lambda_3$, $\lambda_4$). Three conventional parameters include the wave orbital velocity, wave period, and the BFEV-width and stem spacing-based parameter ($\lambda_1$, $\lambda_2$), and the ratio of stem spacing to wave height ($\lambda_3$). The newly proposed parameter ($\lambda_4$) is the ratio of gravity to wave orbital acceleration, which is significantly positively related to the wave height attenuation. A multiple linear regression formula for $C_t$ based on these four parameters is obtained with a high correlation coefficient of 0.958. This study is expected to supplement the wave attenuation data of this new breakwater and provide fundamental theory for the design and construction of the BFEV.

**Keywords:** breakwater of floating emergent vegetation; wave attenuation; wave transmitted coefficient; multiple linear regression

## 1. Introduction

Wave erosion has endangered the stability of the banks of rivers, lakes, coasts, and marine areas for a long time [1,2]. Further, the possible wave resonance would increase wave load and induce rapid growth of the motion response of coastal and offshore engineering structures [3–6], which would cause the failure of these structures. The breakwater of floating emergent vegetation (BFEV) has been used as a new nature-based breakwater to prevent water banks from wave erosion, and its further application potential to the protection of engineering structures is huge due to its predominant ecological benefit. Compared with various hard breakwaters of concrete, stone, or metal, the BFEV is more habitat-adaptive for rare birds, more scenery-pleasant, more economy-saving, more construction-convenient, and less polluting. Further, compared with ecological macrophytes including the mangrove forest, the salt marsh, and the water-bed-growing emergent vegetation, the BFEV is more site-adaptive and more exclusive of the limitation of water depth. Therefore, it is beneficial to the resilient and sustainable development of rivers, lakes, coasts, and marine areas and is thus expected to be a good alternative to hard breakwaters.

The vegetative strip, including kelp farms, salt marsh, mangrove forests, reed or reed-like emergent macrophyte aquatic vegetation, and even the novel Bragg breakwater, has

been found to significantly affect the dissipation of wave energy and thus the attenuation of wave height at certain wave and vegetation conditions [7–11]. Zhu et al. [12] found in a physical experimental case that over a kelp farm, the wave energy is dissipated up to 29–43% at the incident wave height of 1 m and the wave period of 6 s. Van Rooijen et al. [13] reported in experiments that the submerged vegetation canopy attenuated the wave height by up to 40% when the wave orbital excursion length was greater than the characteristic length of canopy drags. Huang et al. [14] found in a series of indoor experiments that the rigid emergent vegetation of various lengths and porosities caused the solidary wave height to attenuate by 18–62%. Cassidy and Tomiczek [15] observed in a 1:10 geometric scale physical model that the wave heights were attenuated by up to 65% in the mangrove forest. Cao et al. [16] investigated the mangrove field on the south China sea coast and found that wave attenuation occurred over the 100 m width of the mangrove. Massel [17] computed that over 50 m wide in a dense mangrove forest, the wave height was attenuated even by 99%. Zhou et al. [18] observed that the mangrove field caused the attenuation of wave height by nearly 58.33% over a transport distance of 275 m and approximately 80% over a transport distance of 1000 m.

Wave attenuation has been observed to be closely dependent on multiple vegetation characteristics. Cassidy and Tomiczek [15] observed that the attenuation increased with vegetation distribution density and the row width of the mangrove forest. Augustin et al. [19] conducted 46 experimental case tests involving various stem densities and wave properties under emergent and near-emergent vegetation conditions. They found the wave height attenuated by 10.1–24.9% over the first 3 m wide of vegetation with higher density and then further attenuated by 5.7–17.4% over the last 3 m wide with lower density. They also found that emergent vegetation led to higher wave attenuation than near-emergent vegetation. Lee et al. [20] experimentally observed the wave attenuation during a storm event and found the density and width of mangroves were positively correlated to the wave height attenuation. They compared roots, trunks, and canopies and found that mangrove roots contributed more to wave height attenuation. Their observation found no statistical differences between mangrove types, incident wave heights, or water levels.

Simultaneously, wave properties are also reported as important factors influencing wave attenuation. Jadhav et al. [21] found that wave attenuation increased with wave height due to increased orbital wave particle velocities. Cao et al. [16] discovered by spectral analysis that wave energy dissipation depended on wave frequency. Sun et al. [22] modeled vegetation with rigid cylinders physically and numerically under 15 wave conditions of various wave heights, wave periods, and wavelengths, establishing that wave energy dissipation decreases with the increase in wave period. Gao et al. [11] found the Bragg breakwater can utilize the Bragg reflection to significantly alleviate harbor resonance and thus protect the coast for the first time. They investigated the influence of incident wave height on floating box moments. They found the ratios of the second-order components to the corresponding first-order ones around the resonant frequency are normally larger than those at the frequencies far from the resonant frequency, and the larger the incident wave height is, the larger the ratios around the resonant frequency become [4]. Gao et al. [5] further found that the transient gap resonance affected the reflection and energy loss of waves significantly.

The multiple characteristics of vegetation and waves are usually integral to two simple factors, drag force and inertia force, in various analytical and computational models of wave attenuation. Drag force caused by the viscous effect and form drag around the stem is positively related to square velocity and is therefore usually more significant in value compared with inertia force, which is caused by the acceleration of the surrounding fluid and is positively related to a relatively minor accelerated velocity. Consequently, the inertia effect is usually considered a small constant or neglected. Wu and Marsooli [23] numerically simulated long waves over rigid vegetation with various drag coefficients and a constant inertia coefficient of 2.0, finding that the wave height attenuation was positively related to vegetation density. Ozeren et al. [24] experimentally observed the linear and regular waves.



They derived the drag coefficient by attributing the wave-damping effect of vegetation only to drag force and thus neglecting inertia force.

Vuik et al. [25] (cited by Veelen et al. [26]) found that in salt marsh flows, the drag coefficient ranged from 0.13 to 5.75, differing by a factor of 44. This suggested the influencing factors of drag force were complex. Zhou et al. [18] calculated the wave attenuation by integrating the effects of vegetation into a drag coefficient, finding that young trees of nearly 0.55 m high were more effective in attenuating wave energy than the stem part of grown trees of nearly 1.2 m high. This suggested that the variations in the inundation of trees induced by water level fluctuations might affect wave damping. Van Veelen et al. [26] found that the best fit for the drag coefficient is a function of the *KC* number, and the velocity attenuation inside the vegetation is a function of the ratio of wave excursion to stem spacing and the ratio of stem spacing to stem diameter. Different exponents and constants under rigid and flexible vegetation conditions were successfully fitted for the analytical expression of the drag coefficient, including the *KC* number proposed by Kobayashi et al. [27]. Jadhav et al. [28] confirmed an inverse relationship between the *KC* number and the drag coefficient based on measurements in a Spartina alterniflora marsh. Ozeren et al. [24] found the drag coefficient was dependent on the *KC* number when *KC* < 10, increasing with the decrease in *KC* value.

The complexity of vegetated waves causes uncertainty in estimating drag effects and, thus, in the prediction of energy dissipation and wave height attenuation. Work has been conducted to include more parameters of vegetation and waves to obtain more accurate drag estimation and wave dissipation prediction. Tanino and Nepf [29] found in their experiments that the cross-sectionally averaged drag coefficient decreased with the increase in cylinder Reynolds number and increased with the increase in solid volume fraction, and viscous drag per unit cylinder length is independent of solid volume fraction in the range of 0.15–0.35. Stone and Shen [30] replaced the apparent velocity with a cross-sectional velocity dependent on the stem diameter and the lateral spacing of vegetation stems and obtained a more accurate drag coefficient. He et al. [31] estimated the drag coefficients proposed by Dalrymple et al. [32] and Kobayashi et al. [26] based on 112 sets of indoor experiments and further proposed a semi-empirical analytical equation for wave attenuation. They predicted a transmitted wave coefficient ranging from 0.21 to 0.83 due to vertically non-uniform vegetation properties in the root, stem, and canopy. However, differences in the prediction of wave height attenuation are still present. The dimensionless parameters involving multi-characteristics of vegetation, flow, and wave, which include the stem Reynolds number, Froude number, and vegetation volume fraction, have been introduced and applied successfully. While comparison of numerical models using different drag coefficient formulae based on these three dimensionless parameters showed inconsistent wave height attenuation.

It has been found that some new factors might also be important in the prediction of wave attenuation. Jadhav et al. [28] found that regardless of whether *KC* or *Re* is used, there is a reduction in drag coefficient for increased orbital wave particle velocities associated with higher waves. Gao et al. [4] found that both the vertical and horizontal wave forces on the floating body were closely related to the incident wave height and the resonant wave frequency. Augustin et al. [19] numerically simulated the wave attenuation by integrating the effect of vegetation into a bulk drag coefficient. Interestingly, the ratio of numerical to observed transmittance was lower than 1, at 0.91–0.97 over the first 3 m width of denser-distributed vegetation, while it was higher than 1, reaching 1.01–1.04 over the last 3 m width of sparser vegetation. They explained that this error and bias could result from the inherent error in the empirical equations for friction factor, missing two-horizontal-dimension terms in these equations, or some neglected turbulent or vertically varying physics not captured by the Boussinesq simulation.

The previous study discovered the critical influence of various parameters on wave dissipation. However, a more accurate prediction of wave height attenuation still needs more attempts with new parameters. As a new vegetated breakwater, the BFEV's effect

and efficiency on wave attenuation still need to be revealed. The influencing parameters applied for the previous breakwaters need to be tested, and new parameters are worthy of attempting. The limitation is mainly due to the availability of data. Therefore, aiming at supplementing data, selecting appropriate parameters for the BFEV, and attempting new parameters that might influence wave attenuation significantly, the present study will conduct a series of indoor physical experiments under various BFEV and wave conditions. Based on these experiments, the dependence of wave height attenuation on multiple parameters, including the previous and newly proposed ones, will be analyzed. The results are expected to provide some new parameters for wave height attenuation and provide a fundamental guide for the design and construction of the BFEV in application.

## 2. Theoretical Background and Methodology

Wave dissipation over the vegetation strip and the BFEV is affected by the physical and geometric properties of the vegetation strip, including strip width, vegetation density, and stem spacing, as well as the wave and water features, including incident wave height, wave period, and water depth. Other parameters, including submergence ratio, stem diameter, biomass, and flexibility of the vegetation, remained constant in all the experimental tests, and thus they were found to have no significant relationship with wave height attenuation.

The drag force and inertia force, and thus the wave height attenuation, are under the combined effects of the complex features of vegetation and waves. Based on the previous study and the observed data of this study, four combined dimensionless parameters are found to be monotonically related to the transmitted coefficient of wave and thus are used to predict the wave height attenuation in regressive analysis.

The Keulegan–Carpenter ($KC$) number represents the contrast effect of drag force and inertia force. In this study, the width of the BFEV ($B$) is used in the $KC$ number as the characteristic length of a solid since it is the largest length scale of the BFEV, which is defined as

$$\lambda_1 = KC = \frac{VT}{B} \tag{1}$$

where $\lambda_1$ is the first dimensionless parameter that will be used in the regressive analysis in this study; $V$ is the horizontal orbital velocity of the incident wave; $T$ is the wave period.

As the second largest length scale of the BFEV, the stem spacing also affects the drag force and the inertia force, which composes the second parameter

$$\lambda_2 = \frac{VT}{D} \tag{2}$$

where $\lambda_2$ is the second dimensionless parameter chosen by the authors, which is related to the stem density of the BFEV; $D$ is the stem spacing of the vegetation.

The third dimensionless parameter, the ratio of wave height and the stem spacing of vegetation, is also closely related to wave height attenuation, which is expressed as

$$\lambda_3 = \frac{H}{D} \tag{3}$$

where $H$ is wave height.

And a new dimensionless parameter was found closely related to wave height attenuation, which is defined as

$$\lambda_4 = \frac{g}{(V/T)} = \frac{1}{F_r^2} \frac{VT}{h} \tag{4}$$

where $F_r$ is the wave Froude number (Zhu et al. [12]); $h$ is the water depth in a still state.

The transmission coefficient is expressed as the ratio of transmitted wave height to incident wave height,

$$C_t = \frac{H_t}{H} \tag{5}$$

where $H_t$ is the transmitted wave height.

In this study, the dependent relationships of $C_t$ and the four dimensionless parameters ($\lambda_1$, $\lambda_2$, $\lambda_3$, $\lambda_4$) are represented through physical experiments. Further, a prediction formula for $C_t$ is to be regressed based on these relationships.

## 3. Experimental Setup

Experiments were conducted in a rectangular cross-sectional water flume with two glass side walls in the laboratory of harbor, coastal, and hydraulic engineering at Shanghai Jiaotong University. The flume is 20 m long, 1.0 m wide, and 0.8 m high. A swaying wave generator was installed at the upstream end of the flume to generate two-dimensional regular waves (Figure 1). Three rectangular steel plates with uniformly punched holes were set at a slope of 1:4 at the downstream end of the flume, which were parallel to each other to dissipate wave energy and avoid the reflection effect of waves (Figure 1).

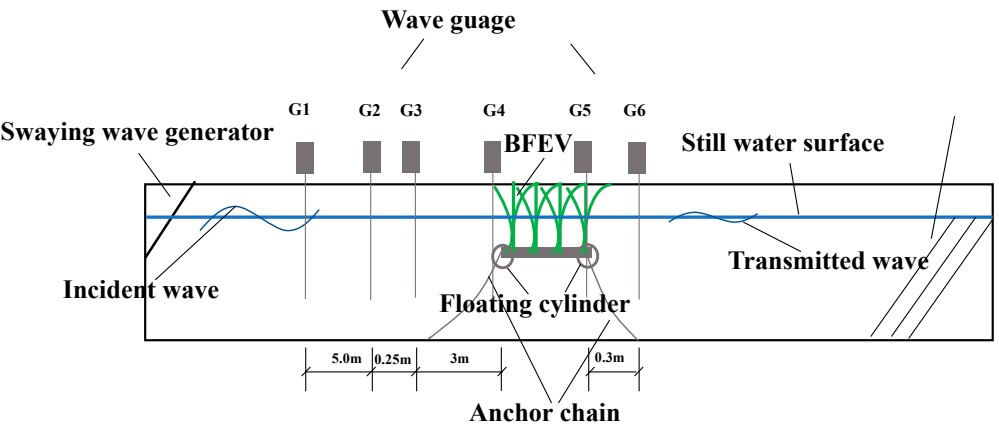

**Figure 1.** Side view of the water flume.

The BFEV model was installed in the middle part, 10 m from the upstream end of the water flume. Two hollow cylinders with a diameter of 5 cm were installed under the BFEV as floaters. Two mooring chains with a stiffness of 1.8 KN/m were designed to anchor the BFEV to meet the requirements of the real application as well (Figure 1). The BFEV emerged from the water surface, with its submerged part deeper than the experimental wave height. Corresponding to the two-dimensional waves in the water flume, the BFEV was installed, occupying the full cross width of the flume (Figure 2). The individual vegetation model used the real foliage branch of a plant with a uniform stem diameter ($d$) of 0.006 m. The scale of the model vegetation diameter to the prototype was selected at 1:10. The height of BFEV was 35 cm, with 20 cm emerging under and 15 cm emerging above the still water surface. The baseboard of the BFEV was 18 cm below the water surface. The modulus of elasticity in the bending of a single vegetation stem was $0.98 \times 10^{10}$ Pa and thus was considered rigid. *Morinda citrifolia* L. was used to model the coastal bush vegetation. Vegetation models were in alignment in both lines and rows (Figure 3). The vegetation models of foliage branches were fixed on a bottom board made of Perspex, and the bottom board was moored on the bottom of the water flume (Figure 4). Both the width of the BFEV ($B$) and the stem spacing ($D$) vary in different experimental cases.

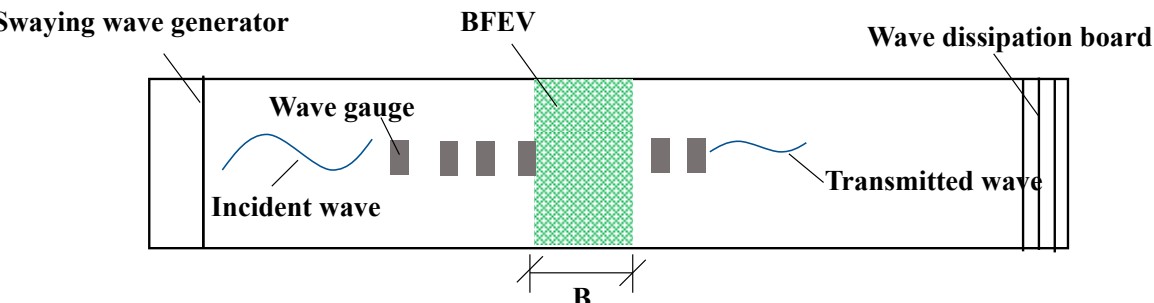

**Figure 2.** Horizontal planer view of the water flume.

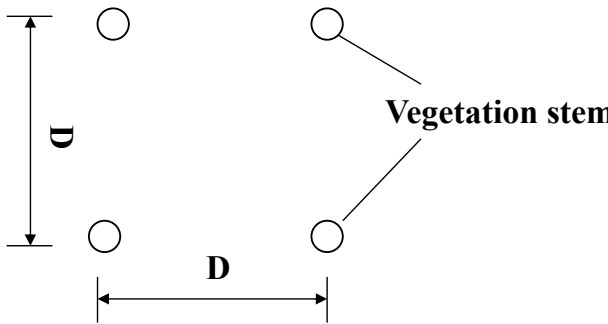

**Figure 3.** Arrangement of vegetation stem.

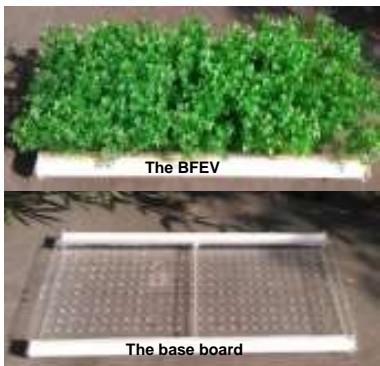

**Figure 4.** Camera photo of the BFEV and the baseboard.

The incident wave height ($H$), wave period ($T$) and horizontal wave orbital velocity ($V$) varied in the experiments, but the water depth was constant at 0.35 m for all the experiments. A total of 312 sets of experimental cases were designed by cross-combination of different BFEV and wave conditions, including five BFEV widths, eight stem spaces, two incident wave heights, four wave periods, and four wave propagating velocities. The parameters of different experimental conditions are summarized in Table 1.

**Table 1.** Parameters of the FBEF and wave.

| $B$ (m) | $D$ (m) | $d$ (m) | $H$ (m) | $T$ (s) | $V$ (m/s) |
|---|---|---|---|---|---|
| 0.45, 0.35, 0.25, 0.20, 0.15 | 0.0214, 0.0237, 0.0268, 0.0306, 0.0355, 0.0428, 0.0538, 0.0713 | 0.006 | 0.06, 0.08 | 0.881 | 1.192 |
| | | | | 0.784 | 1.084 |
| | | | | 0.693 | 1.082 |
| | | | | 0.64 | 1.016 |

The SDA100 system of wave sensors and data collection was used to observe the wave parameters. Six wave gauges were set along the wave direction to obtain the wave height.

Guages 1 to 4 were set 8.35 m, 3.25 m, 3 m, and 0 m upstream of the BFEV, and gauges 5 and 6 were set 0 m and 0.3 m downstream of the BFEV. Gauges 2 and 3 were used to obtain the incident wave height by the two-point separation method of Goda and Suzuki [33]. Gauge 6 was used to obtain the transmitted wave height. Gauges 1, 4, and 5 were used to monitor the variation of wave height along the wave direction.

A control test was conducted to estimate the effects of flume walls, the baseboard, and the installation devices of the BFEV on the wave attenuation. It shows that these effects account for less than 1% of the total attenuation of the wave.

## 4. Results

A control experiment without the floating emergent vegetation was conducted to test the influence of the framework of the BEFV and the water flume, and the results show that their influence on wave attenuation is less than 0.15%, so they are not considered in the analysis.

Among the total experimental set of 312 tests, the wave height attenuation ranges from 22.6% to 71.5%, with the transmitted coefficient $C_t$ ranging from 0.285 to 0.774 (Figures 5–8), which validates the significant effect of the BFEV on wave attenuation. To further understand the influencing mechanism of BFEV on wave attenuation, the dependences of $C_t$ on $\lambda_1$, $\lambda_2$, $\lambda_3$, and $\lambda_4$ are analyzed, respectively. Then, to predict the wave height attenuation over the BFEV, the regression function of $C_t$ is analyzed based on these four dimensionless parameters.

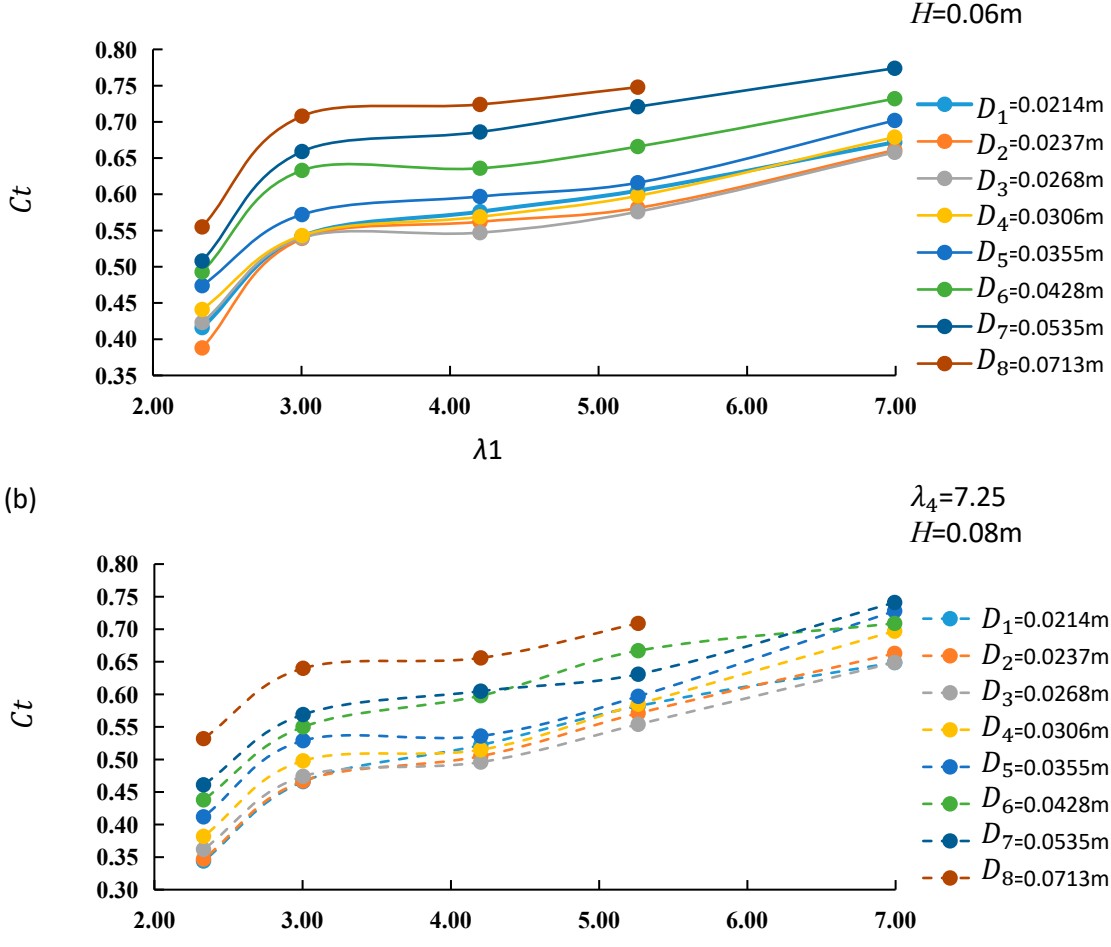

**Figure 5.** *Cont.*

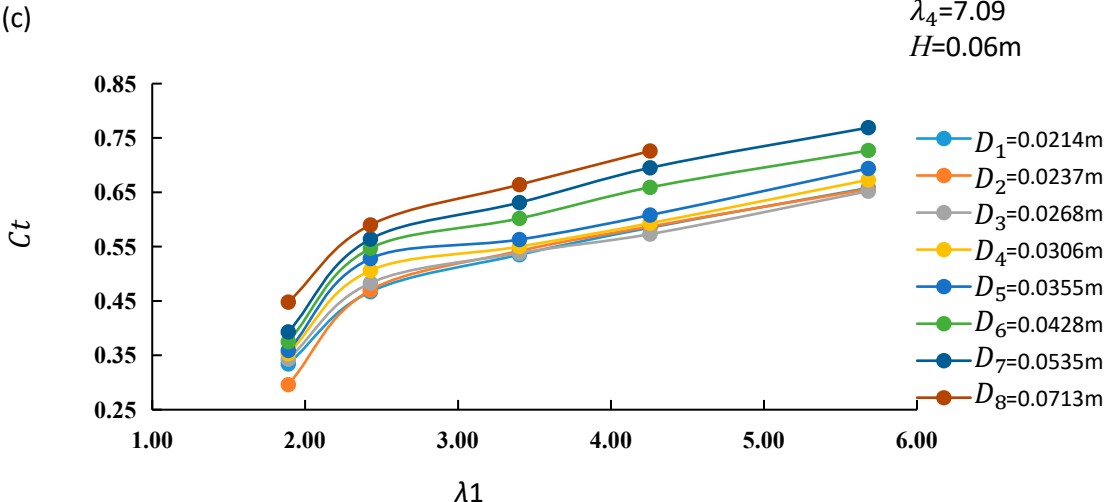

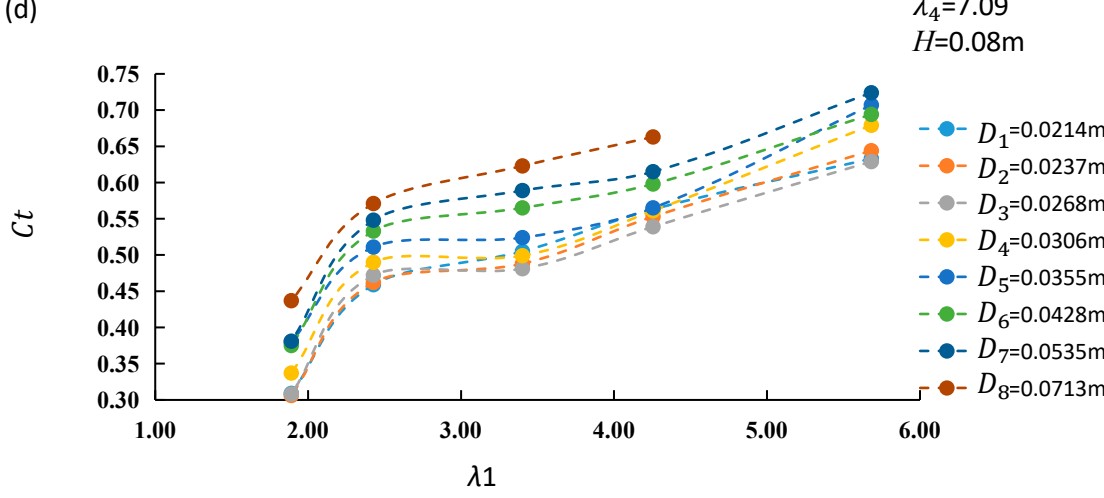

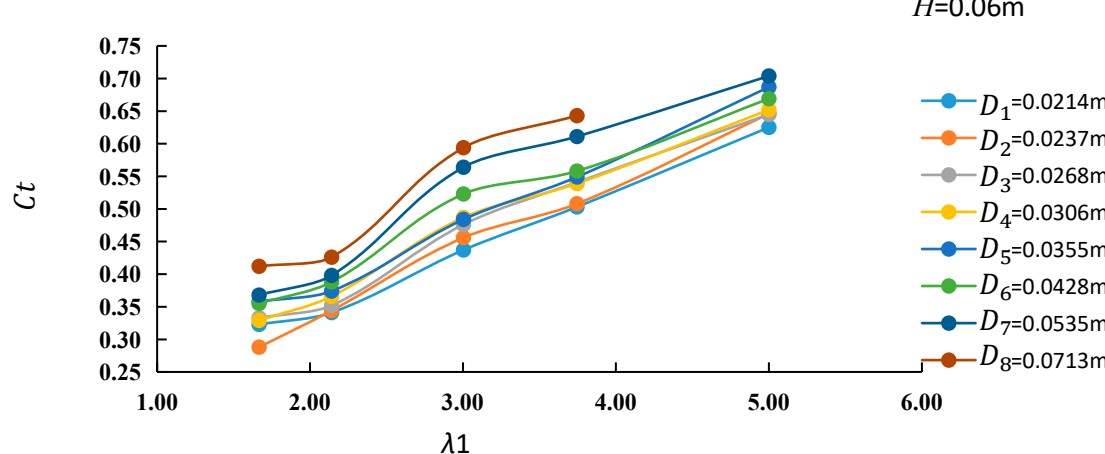

**Figure 5.** *Cont.*

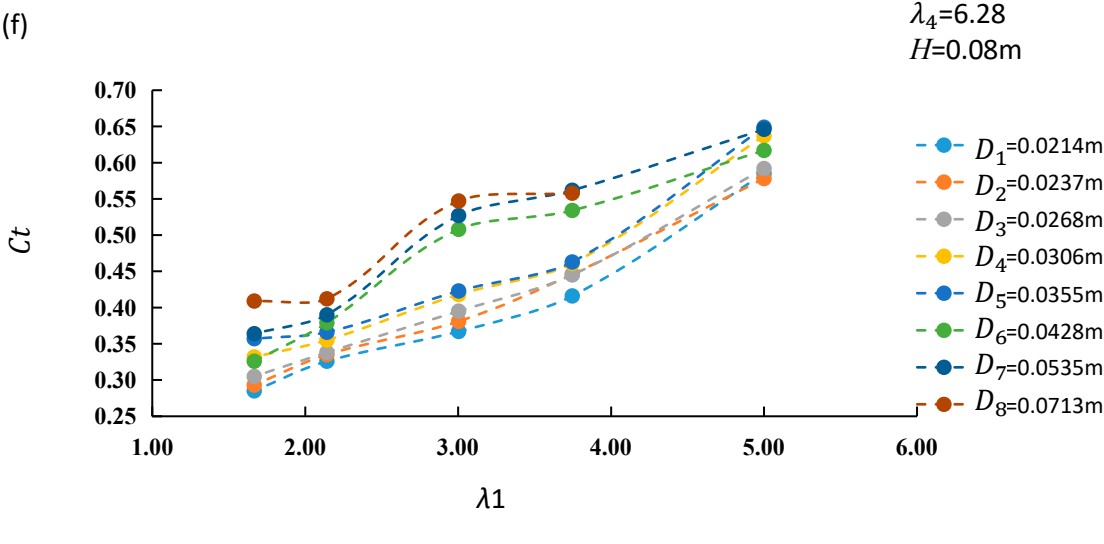

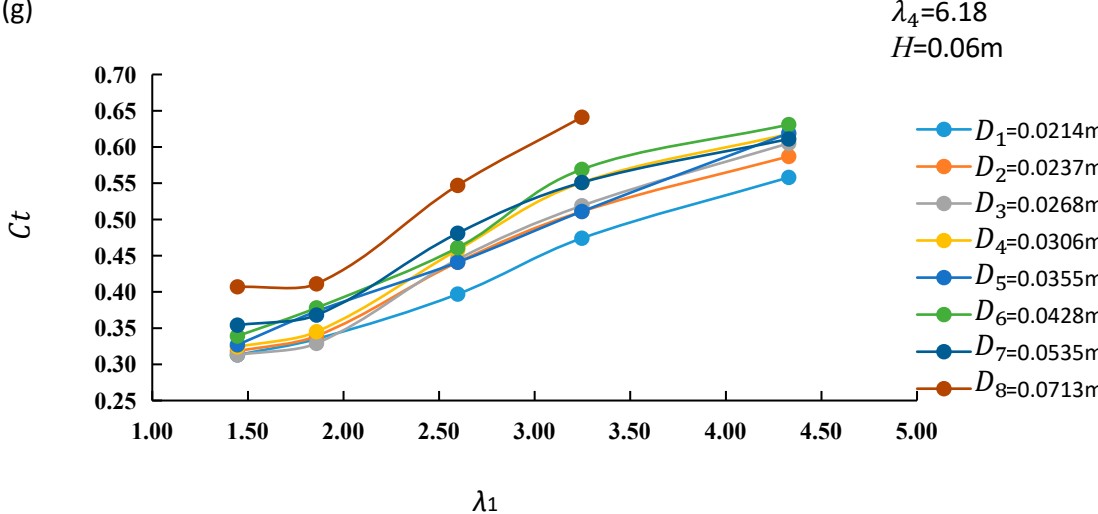

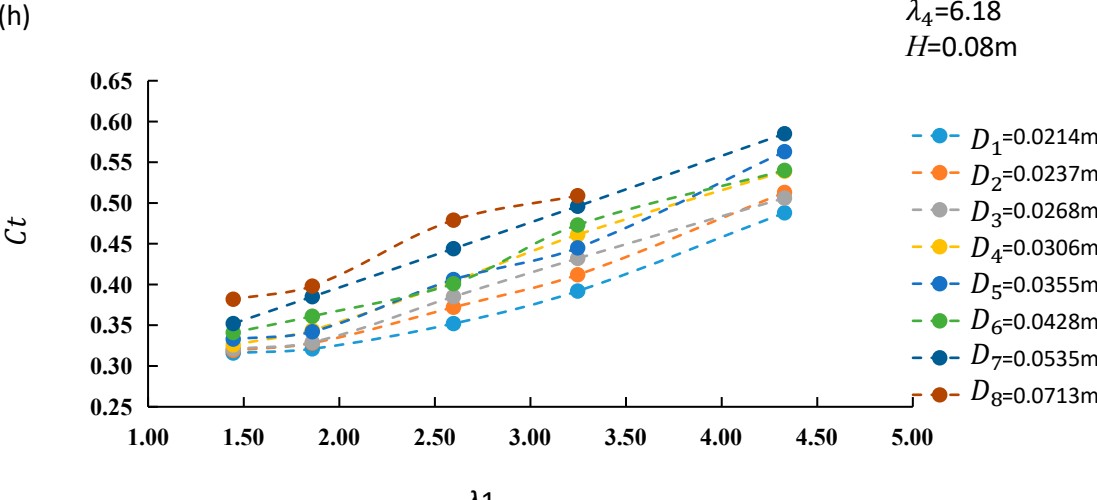

**Figure 5.** Dependence of the wave transmitted coefficient ($C_t$) on the first conventional parameter ($\lambda_1 = KC = \frac{VT}{B}$) under eight stem spacing conditions ($D$ = 0.0214 m, 0.0237 m, 0.0268 m, 0.0306 m, 0.0355 m, 0.0428 m, 0.0535 m and 0.0713 m). Four subfigure pairs (**a**) and (**b**), (**c**) and (**d**), (**e**) and (**f**), and (**g**) and (**h**) are at $\lambda_4 = \frac{g}{(V/T)}$ = 7.25, 7.09, 6.28 and 6.18, respectively, with each subfigure pair corresponding to the wave height $H$ = 0.06 m and 0.08 m.

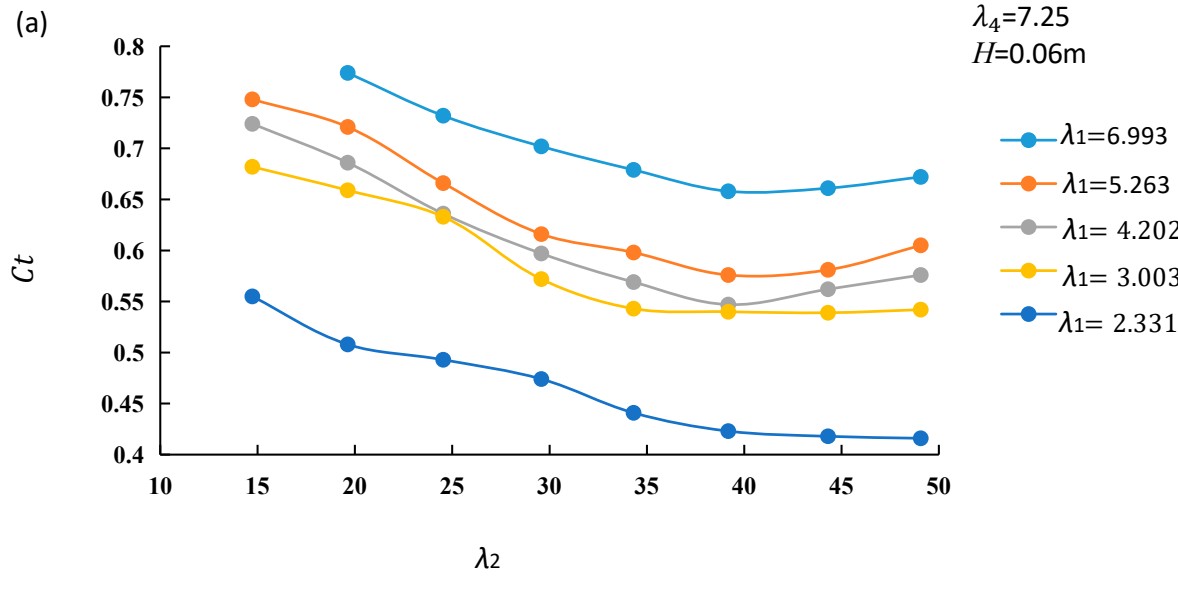

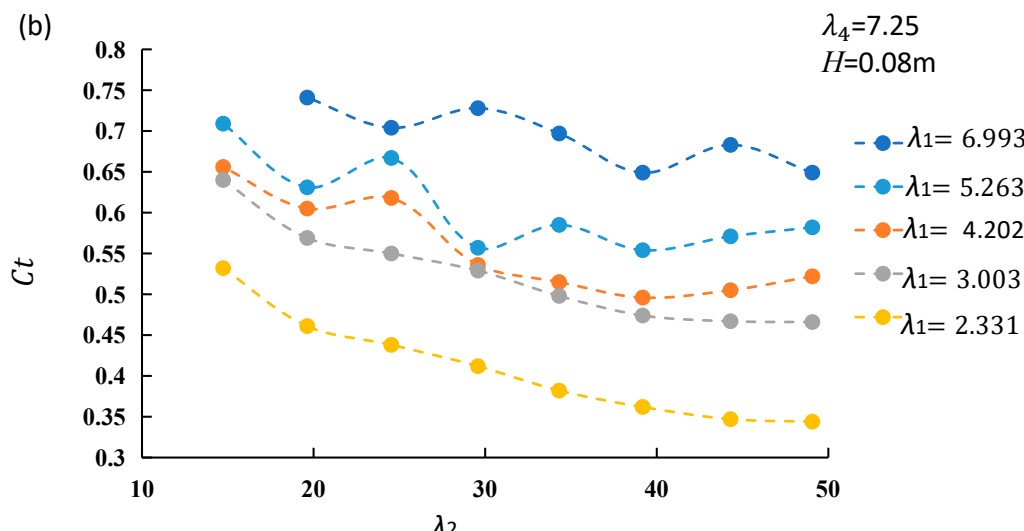

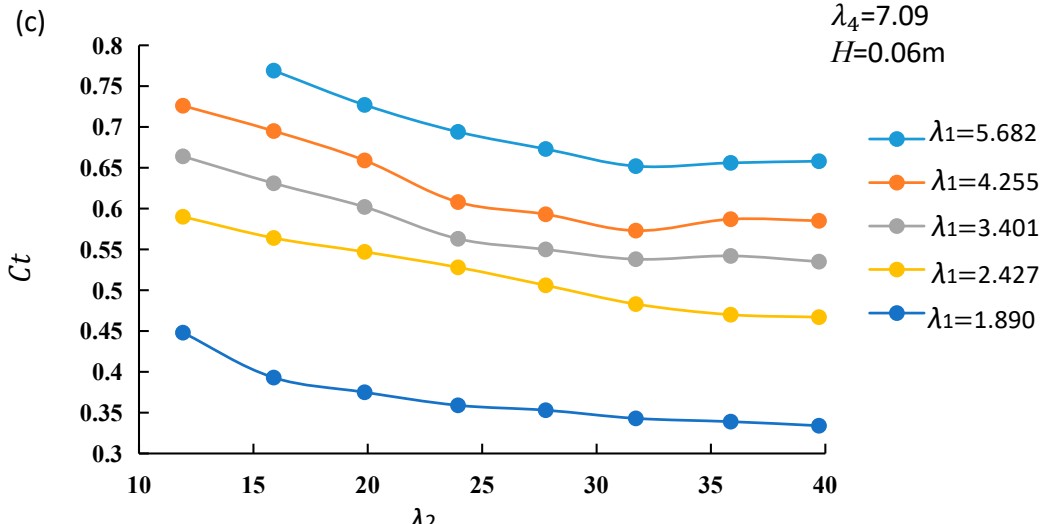

**Figure 6.** *Cont.*

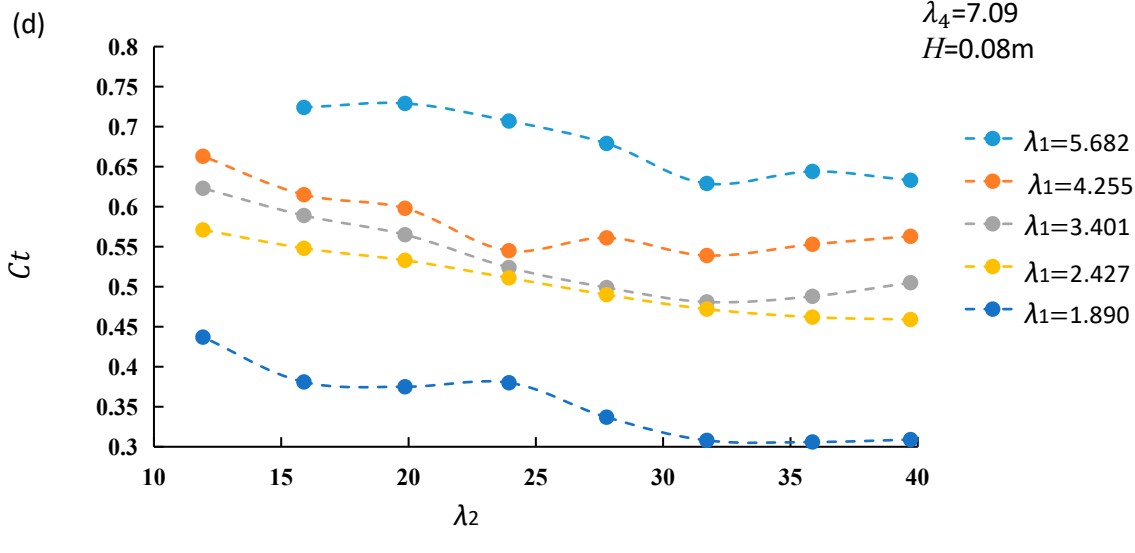

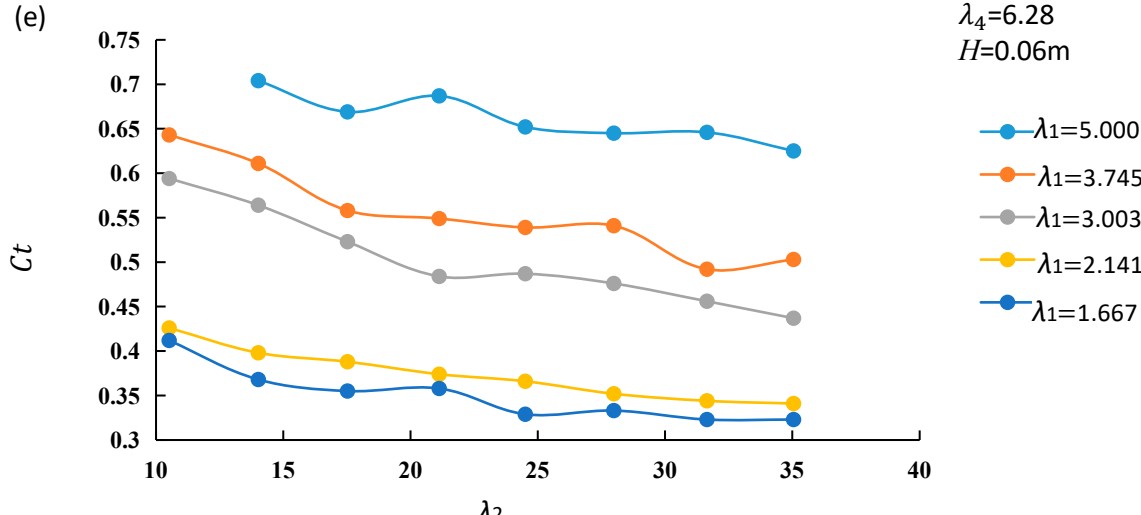

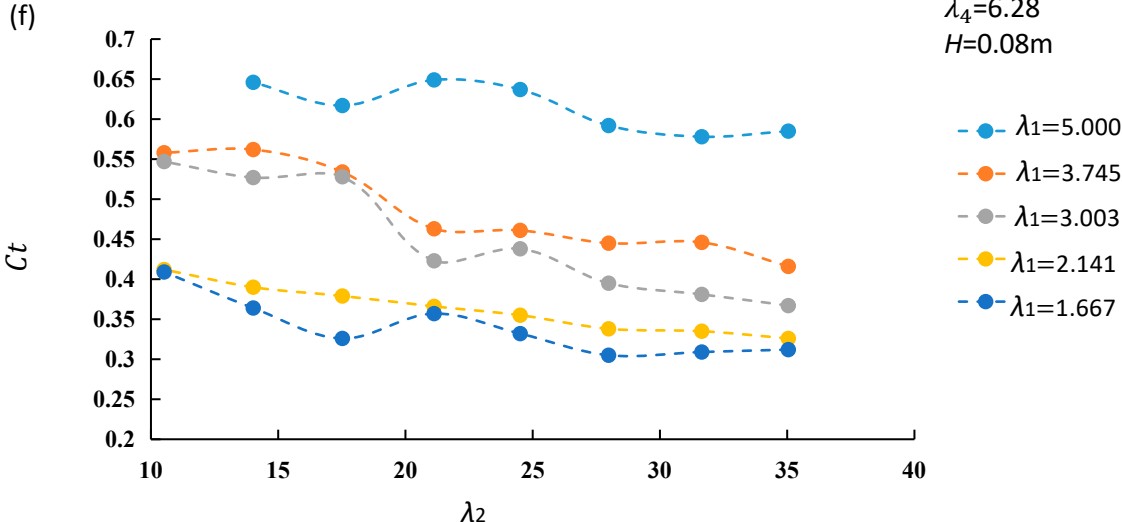

**Figure 6.** *Cont.*

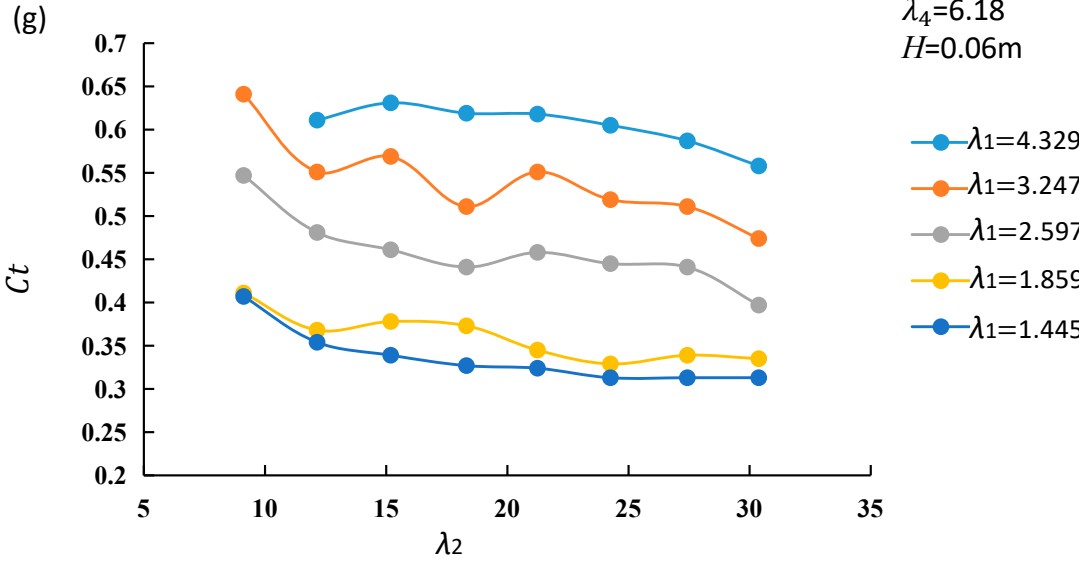

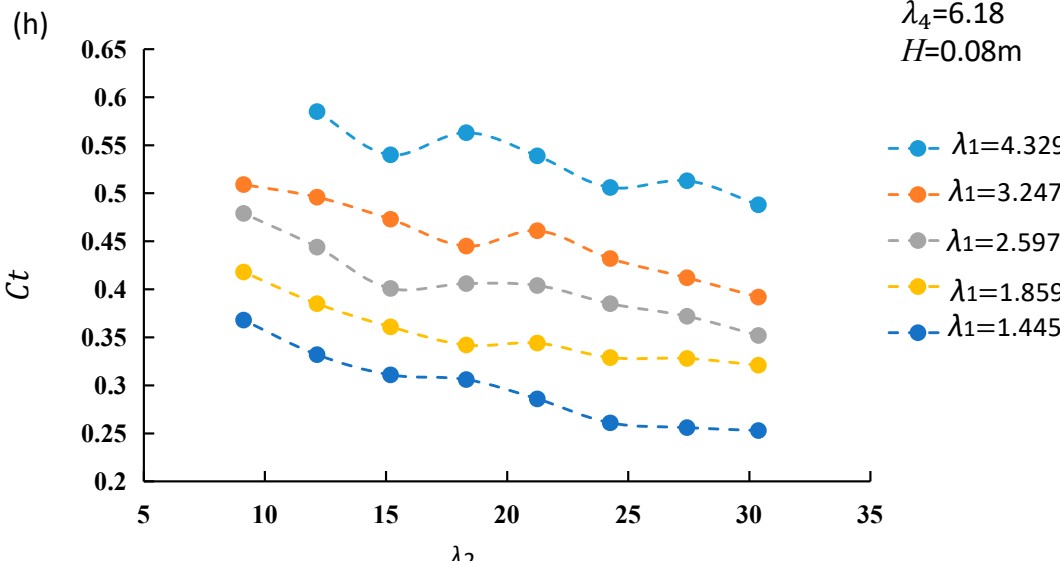

**Figure 6.** Dependence of the wave transmitted coefficient ($C_t$) on the second conventional parameter ($\lambda_2 = \frac{VT}{D}$). Four subfigure pairs (**a**) and (**b**), (**c**) and (**d**), (**e**) and (**f**), and (**g**) and (**h**) are at $\lambda_4 = \frac{g}{(V/T)} = 7.25$, 7.09, 6.28 and 6.18, respectively, with each subfigure pair corresponding to the wave height $H = 0.06$ m and 0.08 m. Subfigure pair (**a**) and (**b**) are at $\lambda_1 = \frac{VT}{B} = 6.993$, 5.263, 4.202, 3.003 and 2.331; Subfigure pair (**c**) and (**d**) are at $\lambda_1 = 5.682$, 4.255, 3.401, 2.427 and 1.890; Subfigure pair (**e**) and (**f**) are at $\lambda_1 = 5.000$, 3.745, 3.003, 2.141 and 1.667; Subfigure pair (**g**) and (**h**) are at $\lambda_1 = 4.329$, 3.247, 2.597, 1.859 and 1.445.

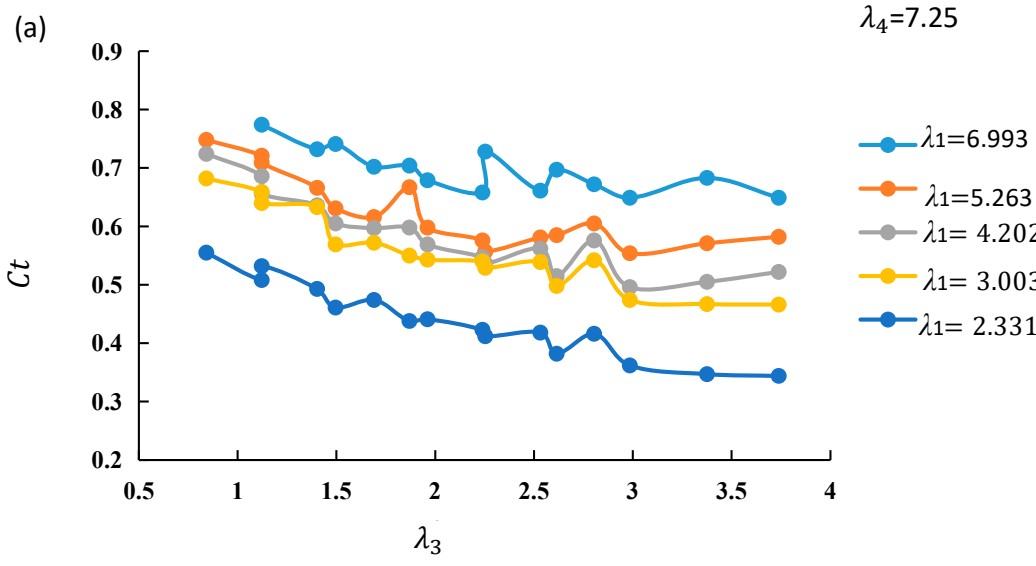

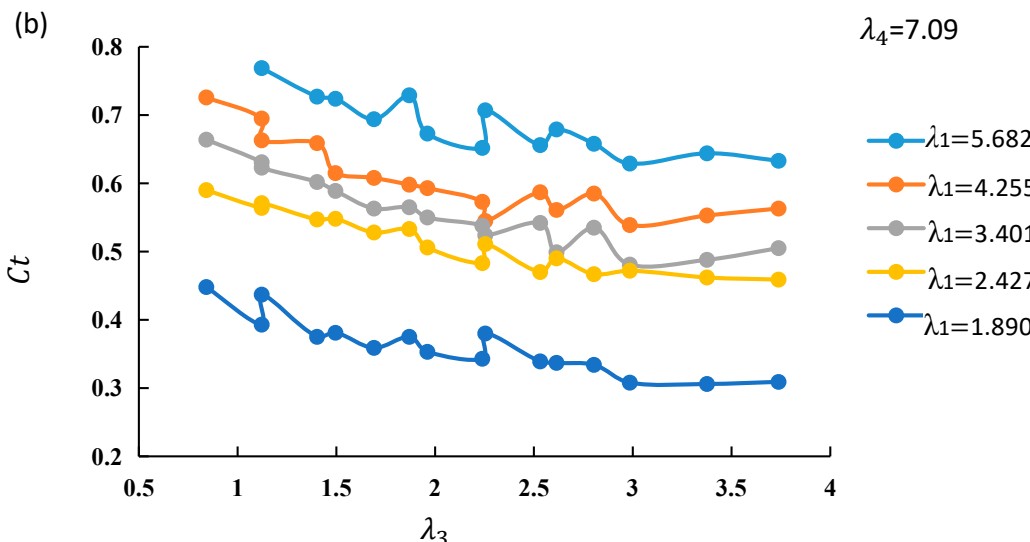

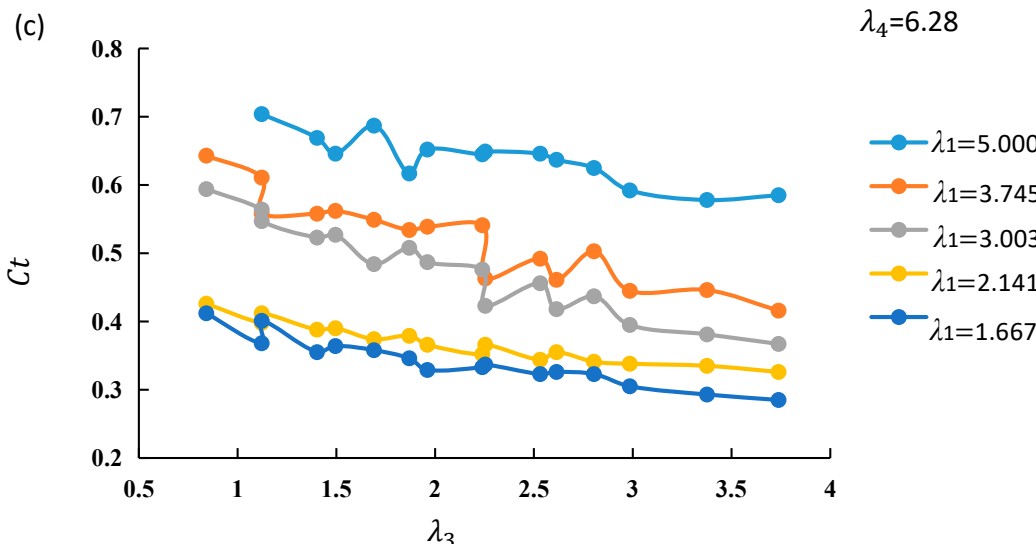

**Figure 7.** *Cont.*

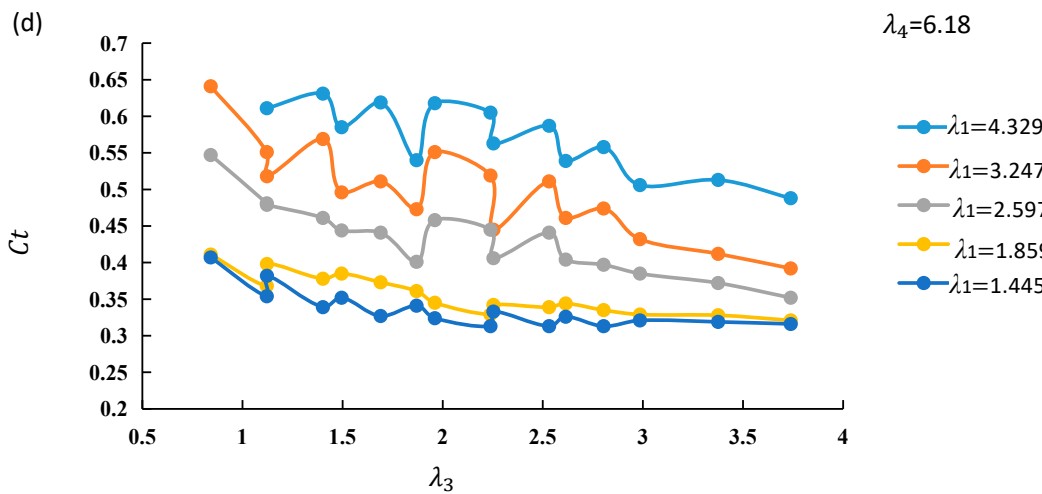

**Figure 7.** Dependence of the wave transmitted coefficient ($C_t$) on the third conventional parameter ($\lambda_3 = \frac{H}{D}$). Subfigure (**a**) is at $\lambda_4 = \frac{g}{(V/T)} = 7.25$, $\lambda_1 = \frac{VT}{B} = 6.993$, 5.263, 4.202, 3.003 and 2.331; Subfigure (**b**) is at $\lambda_4 = 7.09$, $\lambda_1 = 5.682$, 4.255, 3.401, 2.427 and 1.890; Subfigure (**c**) is at $\lambda_4 = 6.28$, $\lambda_1 = 5.000$, 3.745, 3.003, 2.141 and 1.667; Subfigure (**d**) is at $\lambda_4 = 6.18$, $\lambda_1 = 4.329$, 3.247, 2.597, 1.859 and 1.445.

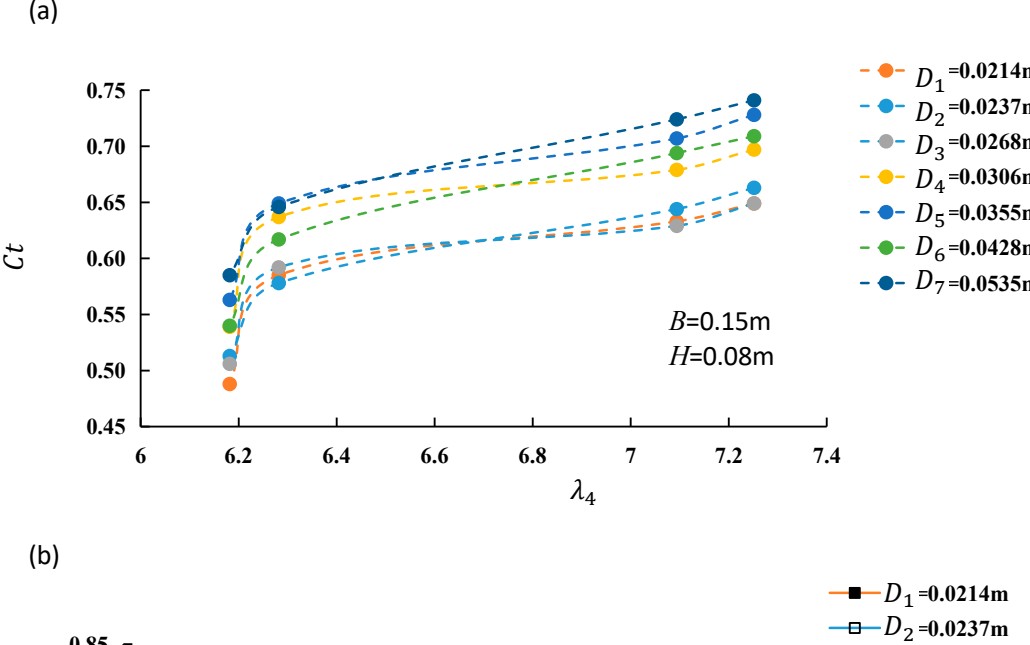

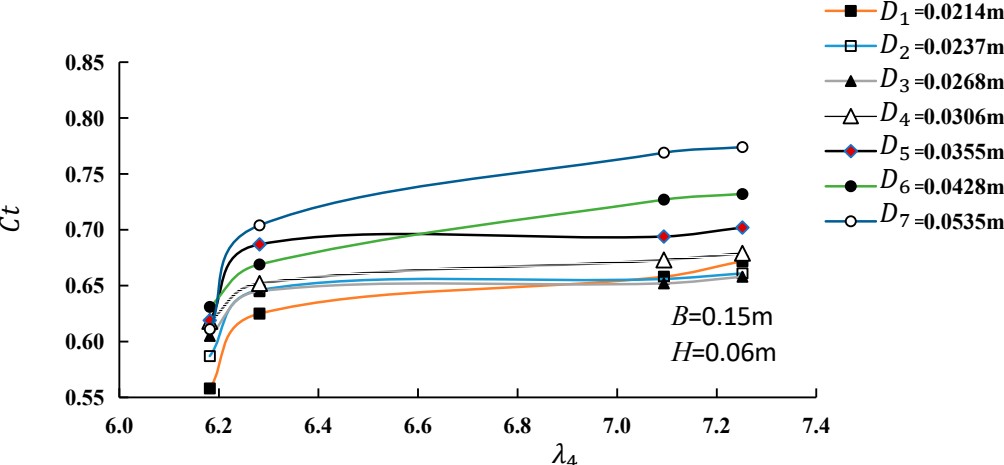

**Figure 8.** *Cont.*

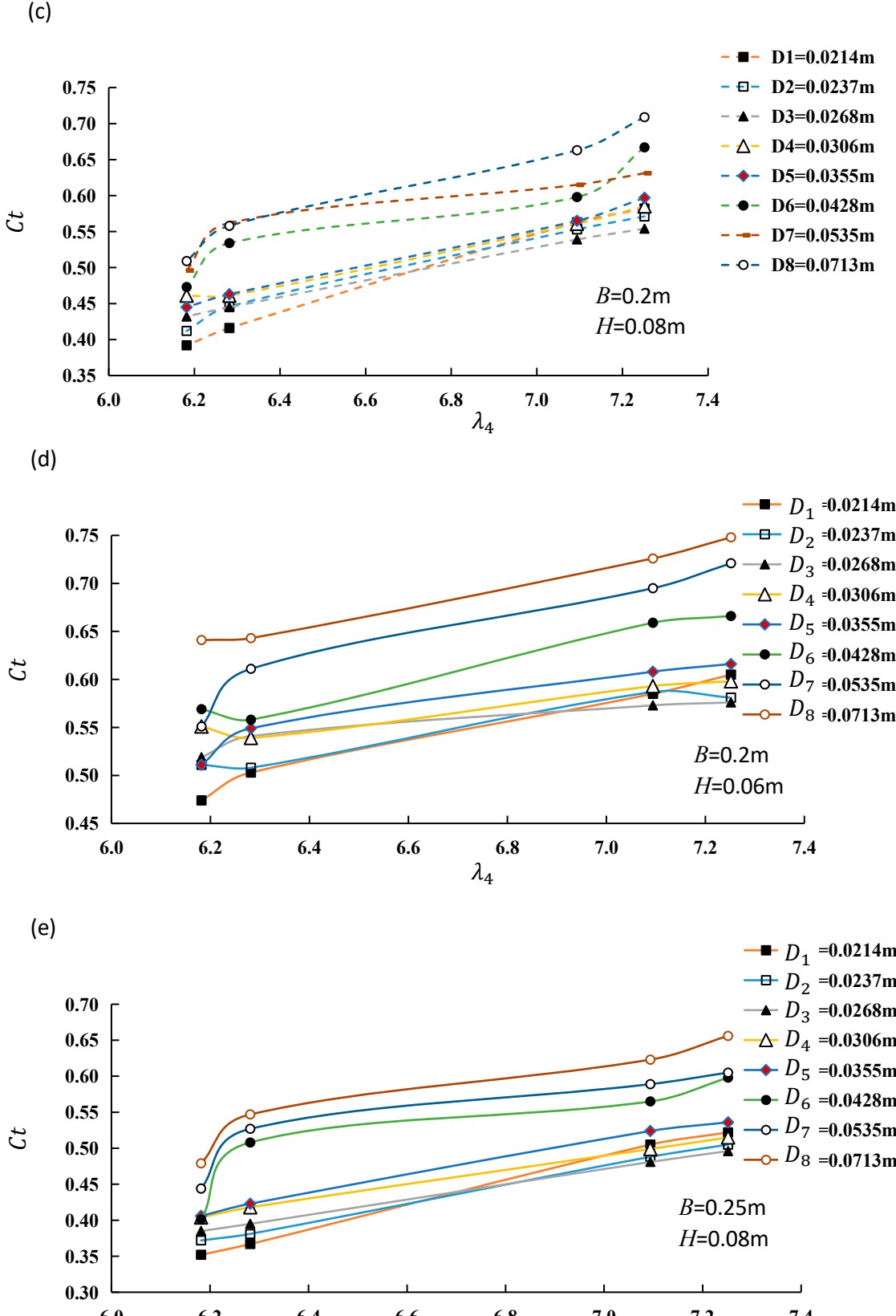

**Figure 8.** *Cont.*

(f)

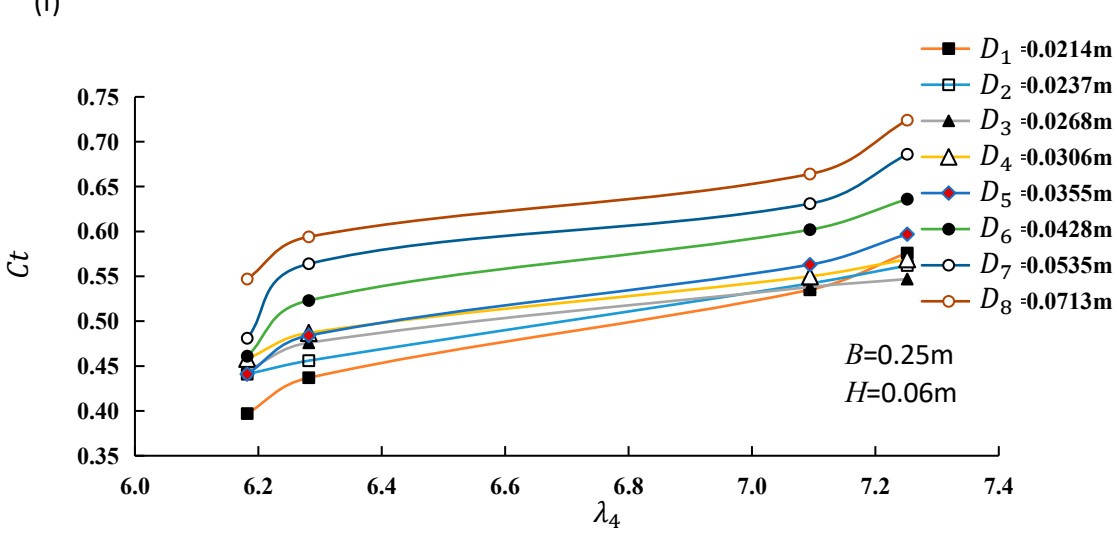

(g)

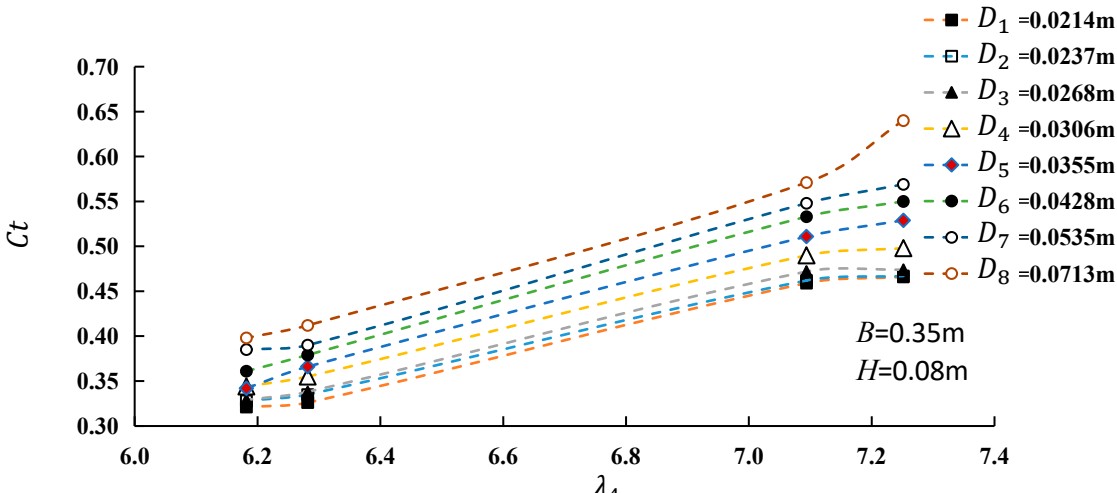

(h)

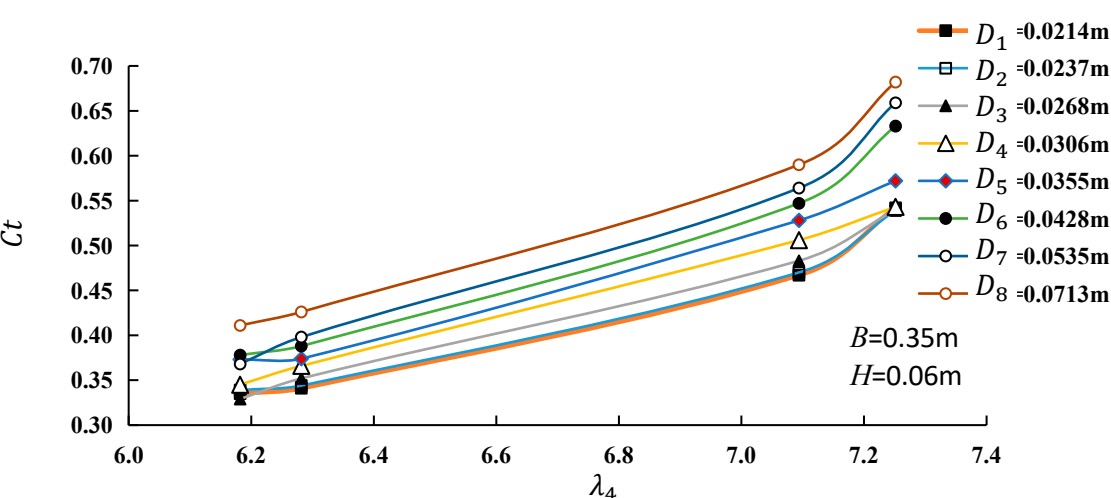

**Figure 8.** *Cont.*

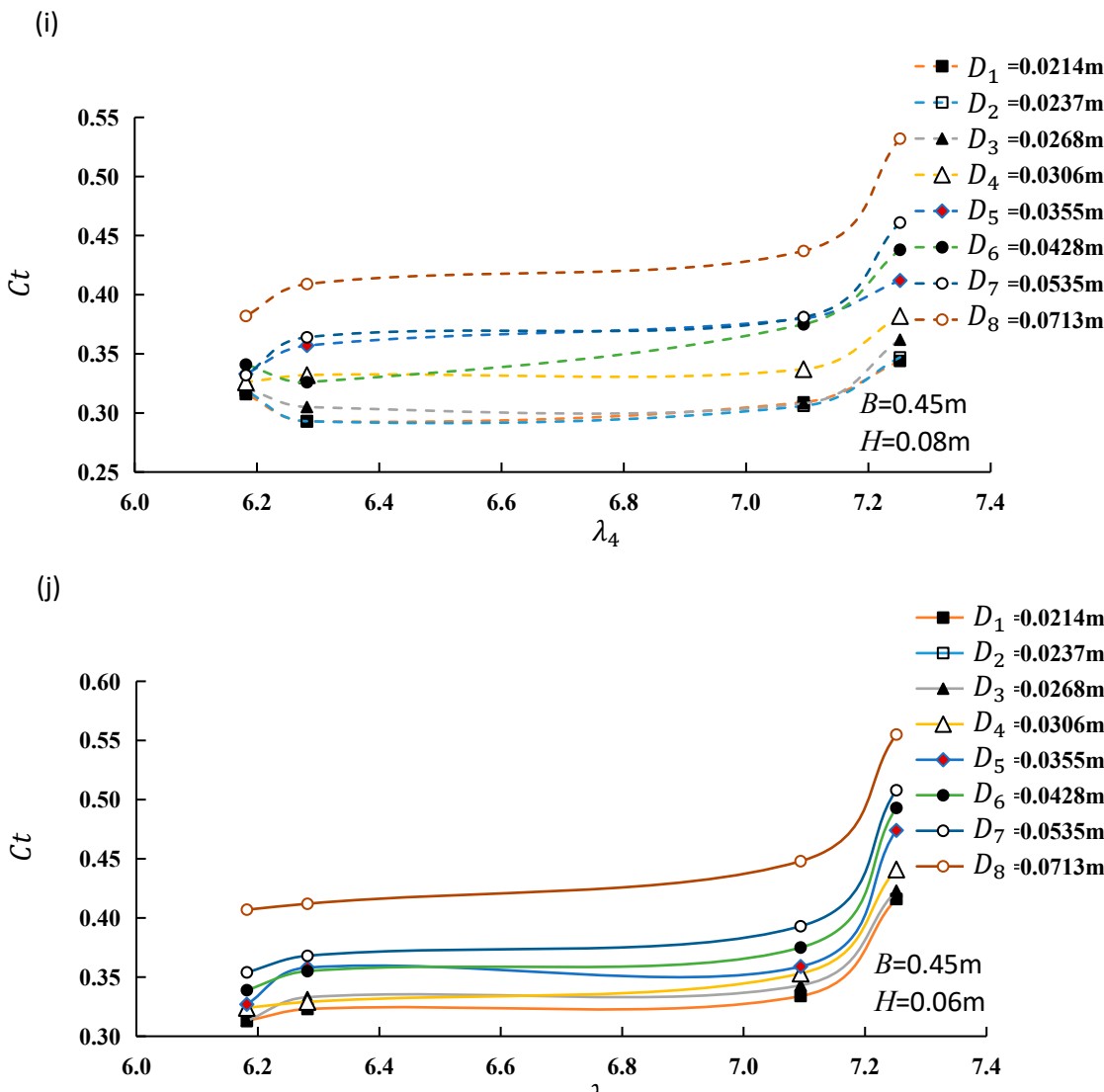

**Figure 8.** Dependence of the wave transmitted coefficient ($C_t$) on the newly proposed parameter ($\lambda_4 = \frac{g}{(V/T)} = \frac{1}{F_r^2}\frac{VT}{h}$) under seven or eight stem spacing $D$ conditions ($D$ = 0.0214 m, 0.0237 m, 0.0268 m, 0.0306 m, 0.0355 m, 0.0428 m, 0.0535 m and 0.0713 m), with Subfigure (**a**) showing seven $D$ conditions and sugbfigures (**b–j**) showing eight $D$ conditions. Five subfigure pairs (**a**) and (**b**), (**c**) and (**d**), (**e**) and (**f**), (**g**) and (**h**), and (**i**) and (**j**) are at the BFEV's length $B$ = 0.15 m, 0.2 m, 0.25 m, 0.35 m and 0.45 m, respectively, with each subfigure pair corresponding to the wave height $H$ = 0.06 m and 0.08 m.

### 4.1. Dependence of $C_t$ on $\lambda_1$

The relationship of $C_t$ and $\lambda_1$ at four $\lambda_4$, two $H$, and eight $D$ conditions is shown in Figure 5a–h with each subfigure under a given combination of $\lambda_4$ and $H$ conditions. Figure 5 shows that, in general, $C_t$ increases with the increase of $\lambda_1$, that is, the wave attenuation decreases with the increase of $\lambda_1$. It is reasonable since both the increase in wavelength and the decrease in the width of BFEV, which increase the value of $\lambda_1$ can lead to a decline in wave attenuation and thus an increase in $C_t$. A comparison of eight lines in each subfigure shows that a smaller $D$ corresponds to a smaller value of $C_t$, indicating denser stem spacing causes more significant wave attenuation.

Comparisons of Figure 5a and b, c and d, e and f, and g and h show that at given $\lambda_4$ and $D$, the value of $C_t$ is smaller at the higher wave height ($H$ = 0.08 m) than at the lower wave height ($H$ = 0.06 m). The difference between these two wave height conditions is

more significant at sparser stem spacing. The largest difference of 0.132 occurs at $\lambda_4 = 6.18$ and $D = 7.13$, with the value of $C_t$ at $H = 0.08$ m being smaller than that of $H = 0.06$ m by 20.6%. This indicates the efficiency of wave attenuation over the BFEV is improved at higher wave heights, and this is more significant under sparser stem spacing conditions.

### 4.2. Dependence of $C_t$ on $\lambda_2$

The relationship of $C_t$ and $\lambda_2$ at four $\lambda_4$, two $H$ and five $\lambda_1$ conditions is shown in Figure 6a–h with each subfigure under a given combination of $\lambda_4$ and $H$ conditions. Figure 6 shows that, in general, $C_t$ decreases with the increase of $\lambda_2$, that is, the wave attenuation increases with the increase of $\lambda_2$. The main reason is that the decrease in stem spacing, which increases the value of $\lambda_2$ can lead to improved wave attenuation and thus a decrease in $C_t$. This suggests that the effect of stem spacing might be relatively important compared with that of the wavelength under the experimental conditions. A comparison of five lines in each subfigure shows that the smaller value of $\lambda_2$ corresponds to a smaller value of $C_t$ and thus leads to a more significant attenuation of the wave, which is in agreement with that of Figure 5.

Comparisons of Figure 6a and b, c and d, e and f, and g and h show that at given $\lambda_4$ and $\lambda_1$, the value of $C_t$ is smaller at the higher wave height ($H = 0.08$ m) than that of the lower wave height ($H = 0.06$ m). The difference ranges from 0.011 to 0.072. The largest difference of 0.072 occurs at $\lambda_4 = 7.25$, $\lambda_1 = 2.33$ and $\lambda_2 = 49.06$, with the value of $C_t$ at $H = 0.08$ m being smaller than that of $H = 0.06$ m by 17.3% (Figure 6a,b). This indicates the efficiency of wave attenuation over the BFEV is improved at the higher wave height condition, which is consistent with that of Figure 5.

### 4.3. Dependence of $C_t$ on $\lambda_3$

The relationship of $C_t$ and $\lambda_3$ at four $\lambda_4$, two $H$ and five $\lambda_1$ conditions is shown in Figure 7a–d, with each subfigure under a given combination of $\lambda_4$ and $H$ conditions. Figure 7 shows that, in general, $C_t$ decreases with the increase of $\lambda_3$, indicating the wave attenuation is improved with the increase of $\lambda_3$. The main reason is that the increase in wave height and the decrease in stem spacing, which increase the value of $\lambda_3$ can lead to an improvement in wave attenuation and thus a decrease in $C_t$. A comparison of five lines in each subfigure shows that the smaller value of $\lambda_1$ corresponds to the smaller value of $C_t$ and thus leads to a more significant attenuation of the wave, which is in agreement with that of Figures 5 and 6.

### 4.4. Dependence of $C_t$ on $\lambda_4$

The relationship of $C_t$ and $\lambda_4$ at five $B$, two $H$, and eight $D$ conditions is shown in Figure 8a–j with each subfigure under a given combination of $B$ and $H$ conditions. Figure 8 shows that, in general, $C_t$ increases with the increase of $\lambda_4$, which indicates the wave attenuation is improved with the increase of $\lambda_4$. This suggests that the increase in horizontal inertia of waves might cause an improvement in wave attenuation over the BFEV. Comparison of the seven (Figure 8a,b) or eight lines (Figure 8c–j) under different $D$ conditions in each subfigure shows that the value of $C_t$ decreases with the decrease of $D$. This is consistent with that of Figure 5, indicating the wave attenuation over the BFEV is improved under denser vegetation conditions.

In consistency with that of Figures 5–7, comparisons of Figure 8a and b, c and d, e and f, and g and h also show that the incident wave height affects the wave attenuation significantly. The wave attenuating efficiency over the BFEV is improved at the higher wave height ($H = 0.08$ m) than at the lower wave height ($H = 0.06$ m).

### 4.5. Regression Analysis of $C_t$ on $\lambda_1$, $\lambda_2$, $\lambda_3$, and $\lambda_4$

Figures 5–8 indicate that the wave height attenuation is closely dependent on the four dimensionless parameters of $\lambda_1$, $\lambda_2$, $\lambda_3$, and $\lambda_4$ (Equations (1)–(4)). Therefore, a multiple linear regression analysis of $C_t$ on these four predominant parameters is conducted.

Let

$$C_t = f(\lambda_1, \lambda_2, \lambda_3, \lambda_4) \tag{6}$$

where $f$ is a function.

Choose a nonlinear multiplication model for the function $f$, obtaining:

$$C_t = \beta(\lambda_1)^{x_1}(\lambda_2)^{x_2}(\lambda_3)^{x_3}(\lambda_4)^{x_4} \tag{7}$$

where $\beta$, $x_1$, $x_2$, $x_3$ and $x_4$ are regression coefficients of the function $f$.

Then, based on the total set of 312 experimental cases, the regressed formula of $C_t$ is obtained.

$$C_t = 0.133(\lambda_1)^{0.508}(\lambda_2)^{-0.016}(\lambda_3)^{-0.207}(\lambda_4)^{0.488} \tag{8}$$

with the correlation coefficient $R = 0.958$.

The calculated and observed $C_t$ shown in Figure 9 indicate they are in good agreement. The agreement is better at the smaller $C_t$, and the deviation is relative larger at the higher $C_t$. The possible reason might be that the wave attenuation was more affected by the bulk swaying motion of the frame of BFEV when the effect of the four dimensionless parameters was relatively smaller at the higher $C_t$.

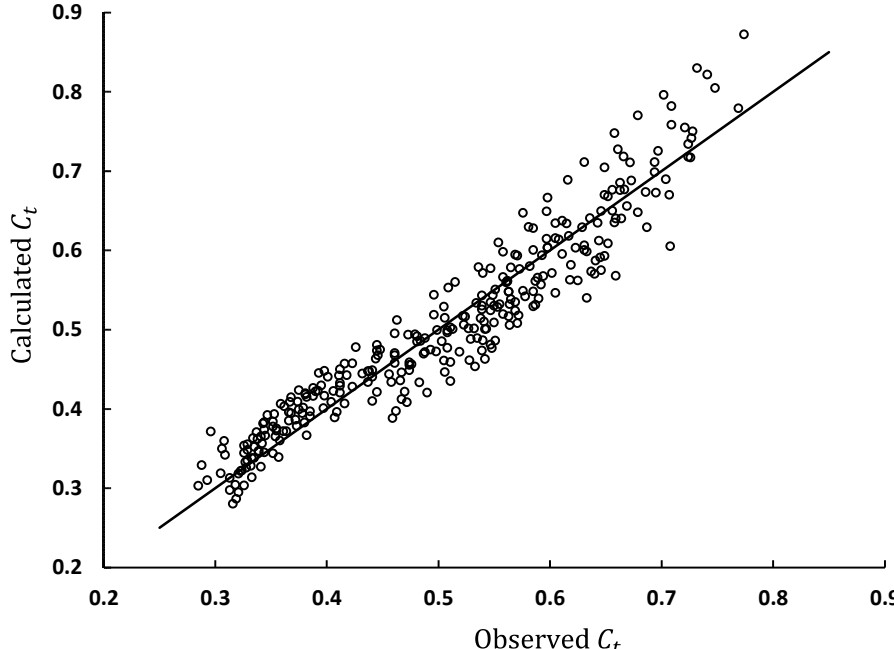

**Figure 9.** The calculated wave transmitted coefficient ($C_t$) versus the observed ones.

## 5. Discussion

Although the regressed Formula (8) agrees with the observed value $C_t$, there is still one point that needs to be noted. The two parameters $\lambda_1$ and $\lambda_2$ have the same form, but they have an inverse effect on wave attenuation and the transmitted coefficient $C_t$. The underlined mechanism is that the width of the BFEV ($B$) is a characteristic length positively related to the resistant force of the BFEV, but the stem spacing ($D$) is negatively related to the resistant force.

For a group of cylinder-conceptualized rigid vegetation stems, the forces, including the drag and inertia effects acting on vegetation, can be estimated by the Morison equation [23,34] ($Fi = \frac{1}{2}\rho C_D N_v A_v U_{vi}\sqrt{U_{vi}U_{vi}} + \rho C_M N_v V_v \frac{\partial U_{vi}}{\partial t}$, where $U_{vi}$ is the apparent velocity acting on the vegetation elements in the $i$th direction, $C_D$ is the drag coefficient, $C_M$ is the inertia coefficient, $N_v$ is the vegetation density defined as the number of vegetation elements per unit horizontal (bed) area, $A_v$ is the projected area defined as the frontal area of a vegetation element projected to the plane normal to the stream-wise flow direction,

and $V_v$ is the volume of a vegetation element). Stone and Shen [30] proposed an alternative drag coefficient $C_{Dm}$ based on the constricted cross-sectional apparent flow velocity $U_{vm}$ with $U_v = U_{vm}(1 - d/D)$. The modified $C_{Dm}$ is more accurate because it is closer to the drag coefficient of a single cylinder and has less variation for a wide range of values for vegetation density, stem size, and cylinder Reynolds number in comparison with $C_D$. The Morison equation and $C_{Dm}$ have been validated by many researchers [15,19,20,23]. In the present study, the width $B$ is positively related to the total projected area of vegetation $A_v$, and the stem spacing $D$ is negatively related to $U_v$ and $N_v$, therefore $\lambda_1$ and $\lambda_2$ have inverse effects on the bulk resistance of the BFEV.

## 6. Conclusions

A series of 312 experimental tests were conducted in an indoor water flume to investigate the effect of the BFEV on wave attenuation, which supplements the data on this new nature-based type of breakwater. Three conventional and one newly proposed dimensionless parameters $\lambda_1$, $\lambda_2$, $\lambda_3$, and $\lambda_4$ are found to have a significant effect on wave height attenuation, with the transmitted coefficient $C_t$ being positively related to $\lambda_1$ and $\lambda_4$, while negatively related to $\lambda_2$ and $\lambda_3$.

A regressed formula of the transmitted coefficient $C_t$ on the four parameters is obtained based on the 312 experimental tests, with a correlation coefficient reaching up to 0.958. The calculated and observed $C_t$ are in good agreement. The relationship between $C_t$ and the four predominant parameters, as well as the regression formula of $C_t$, obtained in this study, are expected to provide fundamental support for the design and construction of the BFEV for bank and structure protection from wave erosion in rivers, lakes, coasts and marine environments.

**Author Contributions:** Conceptualization, Y.L. and G.Y.; methodology, Y.L.; validation, L.X. and D.Z.; formal analysis, Y.L. and D.Z.; investigation, L.X.; resources, G.Y.; data curation, G.Y.; writing—original draft preparation, Y.L.; writing—review and editing, Y.L. and D.Z.; supervision, G.Y.; project administration, G.Y.; funding acquisition, L.X. All authors have read and agreed to the published version of the manuscript.

**Funding:** This research was funded by the National Foundation of the People's Republic of China, grant numbers 51479109 and 51479137.

**Institutional Review Board Statement:** Not applicable.

**Informed Consent Statement:** Not applicable.

**Data Availability Statement:** The data are available at https://doi.org/10.6084/m9.figshare.228329 57 (accessed on 29 June 2023).

**Acknowledgments:** We thank the anonymous reviewers, editors and assistant editors for their constructive comments and suggestions!

**Conflicts of Interest:** The authors declare no conflict of interest.

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
