# Peer review of "Wave Height Attenuation over a Nature-Based Breakwater of Floating Emergent Vegetation"

_sustainability, doi:10.3390/su151410749_

Round 1

Reviewer 1 Report

Major revision is suggested. Please see the attachment.

Extensive editing of English language is required.

Reviewer 2 Report

This study analyses the “ Wave height attenuation over a nature-based breakwater of 2 floating emergent vegetation”. The topic is of high interest, especially in the lack of data in this feild.  The study reports the effect and efficient of the BFEV on wave height attenuation in 14 a set of 312 physical tests in a rectangular indoor water flume. The presentation is clear, the data and statistics robust. However, the Discussion section is not well structured and not derives logically from the Results obtained. On the other hand, this section is to short and do not meet the requirment of the journal.

I think this work could not be published in the journal in current format. However I encourage the authors to take the chance to revise and improve the quality of the mention parts. Overall, I think the study and manuscript fits the journal scope and provides new information.

- My comment refers to the discussion part, which I think could be improved by highliting some parts where comparison with other studies are being done.

- I also see some grammatical issues in the manuscript which I recommend to double check by authors.

- I would recommend to have a comprehensive conclusion at the end of Discussion which could be easily driven from what authors obtained.

- I also see some grammatical issues in the manuscript which I recommend to double check by authors.

Reviewer 3 Report

In the present manuscript investigates the wave height attenuation over a nature-based breakwater of floating emergent vegetation. The article is well-written and organized. The reviewer has the following suggestions and comments that need to be addressed in the manuscript:

1.    The novelty of this study should be highlighted in the Introduction section.   

2.   If possible please compare your results with the earlier published experimental and/or theoretical results. 

3.    Reference style should be rechecked (Year or Number, should not both). 

4.    Some physics in the discussion of the results will be highly appreciated. 

5.    Why bold lines are in the paper (lines 290-291, 296-297)? 

6.    Please write important observations in the conclusion. 

Round 2

Reviewer 1 Report

Major revision is suggested. Please see the attachment.

Moderate editing of English language is required.

Reviewer 2 Report

I think the authors implemented all my  comments and the manuscript could be published in current format. The only minor comment is the figures captions which could be self-descriptive without any referering to the manuscript main body.

Round 3

Reviewer 1 Report

The present version can be accepted.

Minor editing of English language is required.